# Prevalence of and risk factors for adverse events in Alzheimer's patients receiving anti-dementia drugs in at-home care

Hirohisa Imai[1]*, Takuya Hirai[2], Ryosuke Kumazawa[3], Shunsaku Nakagawa[4], Atsushi Yonezawa[4], Kazuo Matsubara[4], Hiroyuki Nakao[5]

1 Department of Medical and Pharmaceutical Community Healthcare, Graduate School of Medicine, The University of Tokyo, Tokyo, Japan, 2 Department of Biostatistics, School of Public Health, Graduate School of Medicine, The University of Tokyo, Tokyo, Japan, 3 Department of Clinical Epidemiology and Health Economics, School of Public Health, The University of Tokyo, Tokyo, Japan, 4 Department of Clinical Pharmacology and Therapeutics, Kyoto University Hospital, Kyoto, Japan, 5 Humanics in Nursing III, Basic Professional Educational Course, Faculty of Nursing, Miyazaki Prefectural Nursing University, Miyazaki, Japan

* imai.hiro@m.u-tokyo.ac.jp

**Data Availability Statement:** All relevant data are within the manuscript.

**Funding:** The authors received no specific funding for this work.

## Abstract

### Objective

The objective of this study was to clarify the types and prevalence of, and the risk factors for, the adverse events that occur in patients receiving anti-dementia drugs.

### Methods

A questionnaire survey was conducted. The respondents were pharmacists who were dispensing anti-dementia drugs. The pharmacists responded to questions about patients who were receiving anti-dementia drugs delivered to them at home by the pharmacists. The survey questions included questions about whether or not the patients experienced adverse reactions to the drugs, about the patients' background characteristics, about the numbers of drugs the patients were taking when the pharmacists first visited the patients at home, and about the pharmacists' assessments of the appropriateness of the use of the anti-dementia drugs.

### Results

Data were collected on 3712 patients from 1673 pharmacies in a nationwide survey. Anti-dementia drugs had been prescribed to 863 of these patients; and 801 (92.8%) of these 863 patients were 75 years of age or older, and. confirmed adverse events occurred in 170 (21%) of these 863 patients. The most common adverse event was excitation/anxiety, at 45.1%. A multivariate analysis found that polypharmacy (10 or more types of drugs per day) (P = 0.030), inappropriate use (P = 0.002), and irregular medication use (P = 0.034) were risk factors.

**Competing interests:** The authors have declared that no competing interests exist.

## Interpretation

In order to avoid adverse events when using anti-dementia drugs, doctors and pharmacists should carefully examine the prescribing of multiple medications, assess the applicability of the use of anti-dementia drugs, and investigate how to best manage patients' drug use.

## Introduction

The number of persons with dementia has been increasing rapidly worldwide in recent years, and has become a global issue associated with far-reaching medical and financial ramifications [1–5]. Although the increasingly elderly nature of the world's population provides the foundation for this trend, other factors include the promotion of government policies [2, 6, 7], increased clinical awareness [3], and drug treatment guidelines [8, 9].

At one time, psychiatric medications were commonly used as drug treatments for dementia [3, 10, 11]. However, in 1996, acetylcholinesterase inhibitors came on the market and began being used [12], and these were followed by NMDA receptor antagonists [13]. The use of these anti-dementia drugs subsequently increased rapidly [3, 14]. The English NICE guidelines initially limited the use of these drugs to severe cases of Alzheimer's dementia [15]. Some patients groups resisted these restrictions, and the drug company that developed donepezil and the distributors in the UK fought a protracted legal battle against the restrictions, eventually winning [16–18]. At present, there are no restrictions on the use of donepezil other than that its use be limited to patients with severe dementia. Acetylcholinesterase inhibitors and NMDA receptor antagonists are widely and routinely prescribed for the treatment of dementia by primary care physicians and other doctors who are not specialists in the treatment of dementia. In both Japan and the UK, donepezil has the leading market share of all of these drugs, followed by memantine and galantamine [19, 20].

Adverse events have been reported in patients using acetylcholinesterase inhibitor and NMDA receptor antagonist anti-dementia drugs. Typical adverse events that occur with the use of acetylcholinesterase inhibitors include gastrointestinal symptoms such as vomiting and diarrhea and psychiatric symptoms such as excitability, insomnia, and hallucinations [21–24]. Patients taking NMDA receptor antagonists sometimes experience adverse events such as dizziness and lightheadedness [23, 24].

Little research has been conducted analyzing adverse event data from patients taking anti-dementia drugs prescribed by physicians for patients with dementia in primary care settings. Specifically, little epidemiological research has been done where the researchers have obtained data on adverse events directly from the clinical setting and thereby gained an accurate understanding of the prevalence of and risk factors for adverse events [25]. One of the reasons for this is that physicians tend to underreport mild adverse events compared to serious adverse events [26, 27]. In addition, most of the anti-dementia drug epidemiology research that has been published has been conducted with funding from drug companies selling anti-dementia drugs, and show low rates for adverse event prevalence [28, 29]. Furthermore, although some information on the frequencies and risk factors for adverse events is available in electronic health records (EHR) [30], there are limits to the accuracy of this information because these are not databases that are intended primarily for the analysis of adverse events. The objective of this study was to clarify the types and frequencies of, and risk factors for, the adverse events that occur in patients taking anti-dementia drugs. We conducted a nationwide survey using a method for obtaining this information from the pharmacists who are handling these patients.

## Methods

### 1) Setting, study design and participants

Setting: In the Japanese at-home care system, pharmacists working at community pharmacies visit patients at their homes and are responsible for obtaining patient consent and handling some of the drug treatment management work, following instructions received from physicians. Pharmacists who visit patients at home obtain patient data (on, for example, the drugs the patients have been prescribed and their disease, medical histories, and test results) as needed from the physicians who are handling the home care of these patients, and then when they visit the patients at their homes they collect information about the patients' symptoms, assess the efficacy of the drugs the patients are taking, collect information about the patient's treatment adherence, and collect information about any signs that could indicate that the patients have experienced drug-induced adverse events. They then determine whether or not this information has been reported to the doctors and if the doctors have been administering the drug therapies appropriately. However, there is not always close and smooth cooperation between the doctors and the pharmacists regarding the drug treatments that are administered in this home care system.

Study design: In this study, we had pharmacists who work for community pharmacies and were visiting patients at home fill out responses to a questionnaire. We conducted a nationwide survey using an original questionnaire on the drug therapies being taken by patients who are being visited at home by these pharmacists. We did not prepare a patient consent form because the data were collected anonymously. We asked these pharmacists to provide information on not more than 5 patients whom the pharmacists were visiting at home, whether or not they were taking anti-dementia drugs. The pharmacists were asked to fill out the questionnaire based on the most recent home visit services they had provided.

Questionnaires: The information about the patients that was obtained using the questionnaire included information about the patients' symptoms and treatment adherence, the appropriateness of the drug therapies, and signs that could indicate the presence of drug-induced adverse events. We conducted this survey by having the pharmacists who were the ones filling out the questionnaires answer questions that had been posted on a web page. The first step in our survey method was to reach out to pharmacists to inform them about the survey and enlist their participation. Specifically, we arranged for the inclusion in a journal that is published monthly by the Japan Pharmaceutical Association of the following insert: "Request for Participation in a Survey About Patient Drug Treatments Encountered in Pharmacists' Home Visit Activities, and Guide for Completing a Web-Based Questionnaire." As the second step, one managing pharmacist from each participating pharmacy (managing pharmacist at the pharmacy) was designated the person who would respond to the questionnaire, and these pharmacists were given access to a specific URL so that they could fill out the questionnaire. The questions were primarily multiple choice questions, although the pharmacists' opinions about their home visit duties were also collected using free response questions. The survey was conducted between January 10, 2018 and March 30, 2018.

### 2) Data collection

The data that we focused on collecting in this survey were data on the frequencies of adverse events, and the risk factors for developing these adverse events, that are experienced by patients who have been prescribed anti-dementia drugs. The survey items also included basic information about the patients (sex, age, level of nursing care required, primary disease, living situation) as well as the types of adverse events being experienced. In addition, in order to

investigate the appropriateness of the drug therapies the patients were receiving, the survey also asked about patients' adherence to the treatments they were receiving when the pharmacists first visited them at home, as well as the numbers of drugs they were taking, and asked the pharmacists to assess the appropriateness of the use of anti-dementia drugs in these patients. In Japan, only 2 types of anti-dementia drugs are covered by health insurance, acetylcholinesterase inhibitors (donepezil, galantamine, and rivastigmine) and NMDA receptor antagonists (memantine), and this survey did not ask pharmacists for the names of the specific anti-dementia drugs their patients were taking.

### 3) Statistical analysis, approval of research ethics review, and role of the funding source

In order to identify the factors associated with the adverse events being experienced by patients who have been prescribed anti-dementia drugs, univariate analyses were performed using a chi-squared test, a chi-squared test (for trend), or Fisher's exact test for the presence or absence of adverse events and for individual variables–that is, patient attributes (sex, age, level of nursing care, living situation), the number of drugs being by the patient when the pharmacist first visited the patient at home, the patient's treatment adherence when the pharmacist first visited the patient at home, and the pharmacist's assessment of anti-dementia drug efficacy. In addition, a multivariate logistic regression analysis was performed using as covariates those patient attributes having a P value of less than 0.20 in the univariate analyses. This study was reviewed and approved by the research ethics committee of the National Institute of Public Health of Japan. In addition, this study was funded by a research grant from the Ministry of Health, Labour and Welfare; no funding of any kind was received from any companies, groups, organizations, etc. with conflicts of interest.

## Results

Data on 3713 patients was received from 1673 pharmacies nationwide. There was no bias towards any particular region. After the data from patients for whom incomplete data had been collected (e.g., sex unknown, age unknown) were excluded, this left 3565 patients from whom data had been collected.

### 1) Basic patient attributes (Table 1)

The mean age of the patients was 80.1 years. There were 1411 male patients, accounting for 40% of the patient population, and 2154 female patients, accounting for 60% of the population. There were 2704 patients who were 75 years of age or older, and who therefore accounted for 79% of the population. The level of nursing care was II in 734 patients (22%) and I in 677 patients (20%). The most common primary disease was dementia, in 868 (26%) of the patients, followed by cardiovascular disease, in 607 (18%) of the patients, cerebral infarction in 410 (12%) of the patients, and cancer, in 276 (8%) of the patients. The most common living situation was living at home alone (1026, or 30%, of the patients), followed by living at home with family (844, or 24%, of the patients), and living in a communal home with caregiver support (806, or 23%, of the patients).

### 2) Attributes of patients who were being prescribed anti-dementia drugs (Table 2)

Of the patients who had been prescribed anti-dementia drugs, 264 (32%) were male and 573 (68%) were female. Patients who were 75 years old or older accounted for 764 (93%) of the

**Table 1. Attributes of patients being handled by the pharmacies surveyed (n = 3713).**

| Parameter | Category | Male (n = 1411) | (%) | Female (n = 2154) | (%) | Overall (n = 3565) | (%) | Overall (n = 3713)a | (%) |
|---|---|---|---|---|---|---|---|---|---|
| Age | Mean | 77.6 | | 81.3 | | 80.1 | | | |
| Age Distribution | < 75 years | 399 | (29.2) | 341 | (16.4) | 740 | (21.5) | 763 | (21.5) |
| | ≥ 75 and < 80 years | 205 | (15.0) | 263 | (12.6) | 468 | (13.6) | 476 | (13.4) |
| | ≥ 80 and < 85 years | 294 | (21.5) | 513 | (24.7) | 807 | (23.4) | 824 | (23.2) |
| | ≥ 85 and < 90 years | 296 | (21.7) | 491 | (23.6) | 787 | (22.8) | 812 | (22.9) |
| | ≥ 90 | 172 | (12.6) | 472 | (22.7) | 644 | (18.7) | 676 | (19.0) |
| | Total | 1366 | (100.0) | 2080 | (100.0) | 3446 | (100.0) | 3551 | (100.0) |
| Nursing care level | Independent | 64 | (4.8) | 75 | (3.7) | 139 | (4.1) | 145 | (4.2) |
| | Needed support grades I-II | 129 | (9.8) | 220 | (10.8) | 349 | (10.4) | 356 | (10.3) |
| | I | 224 | (17.0) | 453 | (22.3) | 677 | (20.2) | 700 | (20.2) |
| | II | 285 | (21.6) | 449 | (22.1) | 734 | (21.9) | 757 | (21.9) |
| | III | 188 | (14.2) | 286 | (14.1) | 474 | (14.1) | 486 | (14.1) |
| | IV | 178 | (13.5) | 225 | (11.1) | 403 | (12.0) | 425 | (12.3) |
| | V | 206 | (15.6) | 273 | (13.4) | 479 | (14.3) | 492 | (14.2) |
| | Application pending | 13 | (1.0) | 10 | (0.5) | 23 | (0.7) | 23 | (0.7) |
| | Unknown | 34 | (2.6) | 39 | (1.9) | 73 | (2.2) | 74 | (2.1) |
| | Total | 1321 | (100.0) | 2030 | (100.0) | 3351 | (100.0) | 3458 | (100.0) |
| Primary Disease | Circulatory disease | 226 | (17.0) | 381 | (18.6) | 607 | (18.0) | 641 | (18.4) |
| | Cerebral infarction | 242 | (18.2) | 168 | (8.2) | 410 | (12.1) | 421 | (12.1) |
| | Osteoarthritis | 24 | (1.8) | 100 | (4.9) | 124 | (3.7) | 126 | (3.6) |
| | Fracture/osteoporosis | 22 | (1.7) | 117 | (5.7) | 139 | (4.1) | 143 | (4.1) |
| | Rheumatoid arthritis | 9 | (0.7) | 48 | (2.3) | 57 | (1.7) | 58 | (1.7) |
| | Amyotrophic lateral sclerosis | 19 | (1.4) | 23 | (1.1) | 42 | (1.2) | 46 | (1.3) |
| | Dementia | 242 | (18.2) | 626 | (30.6) | 868 | (25.7) | 891 | (25.5) |
| | Parkinson's disease | 64 | (4.8) | 91 | (4.4) | 155 | (4.6) | 156 | (4.5) |
| | Other nerve disease | 54 | (4.1) | 78 | (3.8) | 132 | (3.9) | 136 | (3.9) |
| | Chronic respiratory failure | 88 | (6.6) | 53 | (2.6) | 141 | (4.2) | 144 | (4.1) |
| | Cancer | 147 | (11.0) | 129 | (6.3) | 276 | (8.2) | 287 | (8.2) |
| | Renal failure | 28 | (2.1) | 22 | (1.1) | 50 | (1.5) | 53 | (1.5) |
| | Other | 167 | (12.5) | 211 | (10.3) | 378 | (11.2) | 389 | (11.1) |
| | Total | 1332 | (100.0) | 2047 | (100.0) | 3379 | (100.0) | 3491 | (100.0) |
| Living Situation | At home, alone | 352 | (25.6) | 674 | (32.2) | 1026 | (29.6) | 1057 | (29.5) |
| | At home, with spouse | 395 | (28.7) | 269 | (12.9) | 664 | (19.1) | 673 | (18.8) |
| | At home, with family | 341 | (24.8) | 503 | (24.1) | 844 | (24.3) | 878 | (24.5) |
| | At home, with someone other than family | 15 | (1.1) | 15 | (0.7) | 30 | (0.9) | 33 | (0.9) |
| | Communal living with assisted care | 244 | (17.7) | 562 | (26.9) | 806 | (23.2) | 842 | (23.5) |
| | Special nursing home | 30 | (2.2) | 68 | (3.3) | 98 | (2.8) | 102 | (2.8) |
| | Total | 1377 | (100.0) | 2091 | (100.0) | 3468 | (100.0) | 3585 | (100.0) |

a: Including 148 subjects whose sex was unknown

patients who had been prescribed anti-dementia drugs, and patients with a level of nursing care of II accounted for 217 (27%) of these patients, and those with a level of nursing care of I accounted for 205 (26%) of these patients. The most common primary disease among patients prescribed anti-dementia drugs was dementia, in 539 (67%) of the patients, followed by

**Table 2. Attributes of patients being prescribed anti-dementia drugs (n = 863).**

| Parameter | Category | Male (n = 264) | (%) | Female (n = 573) | (%) | Overall (n = 837) | (%) | Overall (n = 863)[a] | (%) |
|---|---|---|---|---|---|---|---|---|---|
| Age | Mean | 23 | (8.8) | 38 | (6.7) | 61 | (7.4) | 62 | (7.3) |
| | < 75 years | 40 | (15.3) | 81 | (14.4) | 121 | (14.7) | 123 | (14.5) |
| | ≥ 75 and < 80 years | 75 | (28.7) | 158 | (28.0) | 233 | (28.2) | 237 | (27.9) |
| | ≥ 80 and < 85 years | 78 | (29.9) | 169 | (30.0) | 247 | (29.9) | 253 | (29.8) |
| | ≥ 85 and < 90 years | 45 | (17.2) | 118 | (20.9) | 163 | (19.8) | 173 | (20.4) |
| | Total | 261 | (100.0) | 564 | (100.0) | 825 | (100.0) | 848 | (100.0) |
| Nursing care level | Independent | 2 | (0.8) | 3 | (0.5) | 5 | (0.6) | 5 | (0.6) |
| | Assistance required levels I-II | 16 | (6.2) | 30 | (5.4) | 45 | (5.6) | 46 | (5.5) |
| | I | 53 | (20.5) | 154 | (27.6) | 205 | (25.6) | 211 | (25.2) |
| | II | 70 | (27.1) | 150 | (26.9) | 217 | (27.1) | 224 | (26.8) |
| | III | 49 | (19.0) | 94 | (16.9) | 139 | (17.4) | 146 | (17.4) |
| | IV | 31 | (12.0) | 56 | (10.1) | 86 | (10.7) | 94 | (11.2) |
| | V | 30 | (11.6) | 63 | (11.3) | 90 | (11.2) | 96 | (11.5) |
| | Application pending | 1 | (0.4) | 2 | (0.4) | 3 | (0.4) | 3 | (0.4) |
| | Unknown | 6 | (2.3) | 5 | (0.9) | 11 | (1.4) | 12 | (1.4) |
| | Total | 258 | (100.0) | 557 | (100.0) | 801 | (100.0) | 837 | (100.0) |
| Primary Disease | Circulatory disease | 29 | (11.2) | 69 | (12.1) | 96 | (11.9) | 105 | (12.3) |
| | Cerebral infarction | 34 | (13.1) | 30 | (5.3) | 62 | (7.7) | 65 | (7.6) |
| | Osteoarthritis | 1 | (0.4) | 11 | (1.9) | 12 | (1.5) | 12 | (1.4) |
| | Fracture/osteoporosis | 3 | (1.2) | 8 | (1.4) | 11 | (1.4) | 13 | (1.5) |
| | Rheumatoid arthritis | 0 | (0.0) | 7 | (1.2) | 6 | (0.7) | 7 | (0.8) |
| | Amyotrophic lateral sclerosis | 1 | (0.4) | 1 | (0.2) | 2 | (0.2) | 2 | (0.2) |
| | Dementia | 153 | (58.8) | 397 | (69.9) | 539 | (66.5) | 561 | (65.9) |
| | Parkinson's disease | 14 | (5.4) | 15 | (2.6) | 28 | (3.5) | 29 | (3.4) |
| | Other nerve disease | 4 | (1.5) | 9 | (1.6) | 13 | (1.6) | 13 | (1.5) |
| | Chronic respiratory failure | 2 | (0.8) | 3 | (0.5) | 4 | (0.5) | 5 | (0.6) |
| | Cancer | 9 | (3.5) | 8 | (1.4) | 17 | (2.1) | 17 | (2.0) |
| | Renal failure | 2 | (0.8) | 3 | (0.5) | 5 | (0.6) | 5 | (0.6) |
| | Other | 8 | (3.1) | 7 | (1.2) | 15 | (1.9) | 17 | (2.0) |
| | Total | 260 | (100.0) | 568 | (100.0) | 810 | (100.0) | 851 | (100.0) |
| Living Situation | At home, alone | 57 | (21.6) | 164 | (28.7) | 217 | (26.6) | 225 | (26.1) |
| | At home, with spouse | 81 | (30.7) | 69 | (12.1) | 148 | (18.2) | 150 | (17.4) |
| | At home, with family | 40 | (15.2) | 83 | (14.5) | 117 | (14.4) | 129 | (15.0) |
| | At home, with someone other than family | 4 | (1.5) | 3 | (0.5) | 5 | (0.6) | 7 | (0.8) |
| | Communal living with assisted care | 77 | (29.2) | 229 | (40.0) | 299 | (36.7) | 319 | (37.0) |
| | Special nursing home | 5 | (1.9) | 24 | (4.2) | 29 | (3.6) | 31 | (3.6) |
| | Total | 264 | (100.0) | 572 | (100.0) | 815 | (100.0) | 861 | (100.0) |

a: Including 26 subjects whose sex was unknown

cardiovascular disease, in 96 (12%), cerebral infarction, in 62 (8%), and Parkinson's disease, in 28 (4%). The most common living situation was living in a communal home with caregiver support, in 299 (37%) of the patients, living at home along, in 217 (27%) of the patients, and living at home with a spouse, in 148 (18%) of the patients.

**Table 3. Disposition of confirmed adverse events in patients prescribed anti-dementia drugs (n = 863) (Multiple answers possible).**

| Adverse Reactions | Male | | Female | | Overall | |
|---|---|---|---|---|---|---|
| | | %[a] | Present, N | %[a] | Present, N | %[a] |
| Overall | 43 | | 122 | | 170[b] | |
| Excitement/insomnia | 23 | 57.5 | 48 | 40.3 | 74 | 45.1 |
| Nausea/vomiting/diarrhea | 12 | 30.0 | 43 | 36.1 | 55 | 33.5 |
| Hallucination/delusion/visual hallucination | 14 | 35.0 | 20 | 16.8 | 35 | 21.3 |
| Poriomania/violent behavior | 6 | 15.0 | 18 | 15.1 | 26 | 15.9 |
| Patch-induced rash | 6 | 15.0 | 10 | 8.4 | 17 | 10.4 |
| Parkinson's-like symptoms | 6 | 15.0 | 8 | 6.7 | 14 | 8.5 |
| Pollakiuria/incontinence | 7 | 17.5 | 5 | 4.2 | 12 | 7.3 |
| Dizziness | 0 | 0.0 | 8 | 6.7 | 8 | 4.9 |
| Bradycardia/arrhythmia | 3 | 7.5 | 2 | 1.7 | 5 | 3.0 |
| Other | 2 | 5.0 | 5 | 4.2 | 7 | 4.3 |

a: Excluding subjects from whom responses were not obtained

b: Including 5 subjects whose sex was unknown

### 3) Presence or absence of adverse events, and disposition of adverse events (Table 3)

In the population of patients who had been prescribed anti-dementia drugs, adverse events were confirmed in 43 (17%) of the males and 122 (23%) of the females, and thus in 170 (21%) of the population prescribed anti-dementia drugs overall. The disposition of adverse events was as follows: excitement/insomnia, in 74 (45%) of patients; nausea/vomiting/diarrhea, in 55 (34%) of patients; hallucinations/delusions/visual hallucinations, in 35 (21%) of patients; and poriomania/violent behavior, n 26 (16%).

### 4) Factors associated with the emergence of adverse events (Table 4)

Table 4 shows the results of a statistical analysis that was performed in order to investigate factors associated with the adverse events identified by the pharmacists. The overall rate of adverse events was 20.9%. Sex, level of nursing care, the number of drugs being taken when the pharmacist first visited the patient at home, the patient's adherence with anti-dementia drug treatment, and the pharmacist's assessment of the applicability of prescribing anti-dementia drugs were identified as marked factors contributing to the development of adverse events. The odds ratio, adjusted for sex and level of nursing care, was 1.51 (95% CI: 1.04–2.19) for a number of drugs when the pharmacist first visited the patient at home of 10 or more, 1.55 (95% CI: 1.03–2.31) for an anti-dementia drug treatment adherence of "forgets to take medication 1–2 times a week," and 2.85 (95% CI: 1.46–5.60) for an assessed applicability of the prescription of an anti-dementia drug of "inappropriate." A high number of drugs being taken when the pharmacist first started visiting the patient at home, an anti-dementia drug treatment adherence of "forgets to take medication 1–2 times a week," and the inappropriate use of an anti-dementia drug prescription were identified as three statistically significant risk factors.

## Discussion

This nationwide survey found that the prevalence of adverse events that were determined by the attending pharmacists to have been caused by anti-dementia drugs in patients receiving anti-dementia drugs exceeded 20%. Although anti-dementia drugs have been widely adopted

**Table 4. Factors associated with adverse event emergence.**

| Factor | N | % | Adverse Event Emergence | | p-value | ORa | 95%CI | p-value |
|---|---|---|---|---|---|---|---|---|
| | | | N | Adverse Event Incidence, % | | | | |
| Overall [b] | 812 | 100.0 | 170 | 20.9 | | | | |
| Patient Characteristic | | | | | | | | |
| Sex | | | | | 0.090[c] | | | |
| Male | 248 | 31.5 | 43 | 17.3 | | | | |
| Female | 539 | 68.5 | 122 | 22.6 | | | | |
| Age | | | | | 0.244[d] | | | |
| Mean | 58 | 7.2 | 10 | 17.2 | | | | |
| < 75 years | 117 | 14.6 | 24 | 20.5 | | | | |
| ≥ 75 and < 80 years | 227 | 28.3 | 44 | 19.4 | | | | |
| ≥ 80 and < 85 years | 235 | 29.3 | 51 | 21.7 | | | | |
| ≥ 85 and < 90 years | 164 | 20.5 | 39 | 23.8 | | | | |
| Nursing care level | | | | | 0.052[c] | | | |
| Independent/Assistance Required/Nursing Care Levels I-II | 454 | 58.7 | 87 | 19.2 | | | | |
| III - V | 320 | 41.3 | 80 | 25.0 | | | | |
| Living Situation | | | | | 0.395[c] | | | |
| At home, alone | 206 | 25.4 | 38 | 18.4 | | | | |
| At home, with spouse | 139 | 17.1 | 26 | 18.7 | | | | |
| Home, living with someone else | 129 | 15.9 | 33 | 25.6 | | | | |
| Communal living with assisted care / Special nursing home | 337 | 41.6 | 73 | 21.7 | | | | |
| Number of drugs being taken at the first pharmacist home visit | | | | | 0.019[c] | | | |
| 0–9 | 530 | 69.6 | 100 | 18.9 | | ref | | |
| 10 - | 231 | 30.4 | 61 | 26.4 | | 1.51 | (1.04, 2.19) | 0.030 |
| Treatment adherence at first pharmacist home visit | | | | | 0.064[c] | | | |
| Not adherent/only taking medication 1–2 times a week | 167 | 21.2 | 27 | 16.2 | 0.562[d] | 0.91 | (0.55, 1.51) | 0.717 |
| Forgets to take medication 1–2 times a week | 230 | 29.2 | 59 | 25.7 | | 1.55 | (1.03, 2.31) | 0.034 |
| Takes medication as directed | 390 | 49.6 | 79 | 20.3 | | ref | | |
| Appropriateness of anti-dementia drug therapy | | | | | 0.009[e] | | | |
| Determined to be not appropriate | 42 | 5.3 | 16 | 38.1 | | 2.85 | (1.46, 5.60) | 0.002 |
| Determined to be appropriate | 753 | 94.7 | 147 | 19.5 | | ref | | |

a: adjusted for sex and level of nursing care

b: Including subjects from whom responses were not obtained for each factor

c: Chi-square test

d: p for trend

e: Fisher's exact test

since they came onto the market, and are now a commonly encountered drug therapy, little research has been done to enable an accurate understanding of the prevalence of adverse events occurring in patients receiving such therapy. The reports that have been published to date include reports of the results of postmarketing surveillance [28, 29], a report based on HER data [30], and a report from a relatively small-scale observational study [31]. These were all studies that were funded by drug companies, and it is possible that they underestimate the adverse events that occurred because of, for example, reporting bias. Our study, however, was conducted using public research funds, and the data were collected using a unique methodology that makes it possible for pharmacists, who are drug product specialists, to evaluate whether or not adverse events occurred. Our study found a higher incidence of adverse events

than the studies that have been conducted to date. Our results suggest that adverse events resulting from anti-dementia drug therapy are more common than has been acknowledged to date.

The most common adverse event in both males and females was excitement/insomnia, occurring in close to half (45.1%) of patients. Poriomania/violent behavior occurred in 21.3%. It was clear, in other words, that excessive mental excitement and uncontrollable physical movement is a frequent and marked adverse event in anti-dementia drug therapy. There is a previous report stating that excitement/insomnia associated with adverse events in anti-dementia drug therapy suggests that the dose being taken is not appropriate [32]. Because acetylcholinesterase inhibitors are metabolized by the liver and NMDA receptor antagonists are excreted renally, in elderly patients in whom these functions are reduced, it is likely that the blood concentrations that are optimal for therapy could be exceeded, resulting in mental excitement. The patients who were studied in our research had an average age of more than 80 years, and had reduced hepatic and renal functions [33]. It is possible that these patients' attending physicians imprudently prescribed anti-dementia drugs without regularly assessing the patients' renal and hepatic function, leading to the patients having excessively high blood levels of the drugs, and thus in the frequent emergence of adverse events.

Although there was no significant difference between the sexes in the incidence of adverse events (P = 0.090), they were more common in female patients. Although there are few reports on differences between the sexes in the incidence of adverse events [34], it is known that females generally experience more adverse events in drug therapy [35]. This is attributed to the fact that women more often take doses that are too high, and to the pharmacokinetics and/or pharmacodynamics. However, it was thought that the reason that the incidence of adverse events was higher in females in this study was that in the distribution by age, although the difference was not significant (P = 0.153), there were more elderly patients among the females (of the 410 patients who were 85 years old or older, 47.1% were male and 50.9% were female).

The nursing care levels described in this study (I through V) were assigned primarily depending on the patient's level of physical ability (freedom in performing activities of daily living), with a level of nursing care of III or above indicating that the patient has difficulty in performing activities of daily living. Patients with a greater need for nursing care were more likely to experience adverse events than patients without such a need (P = 0.052). Such patients may have developed more adverse events because their drug metabolizing functions are generally weakened. Caution should be exercised when using anti-dementia drugs in patients requiring a high level of nursing care.

The mean number of drugs being prescribed to the patients who had been prescribed anti-dementia drugs and had experienced adverse events (n = 170) was 8 (SD = 4.33). It was found that patients who had been taking 10 or more types of drugs per day when first visited at home by the pharmacists, before any pharmacist interventions (231 patients, of whom 61 had developed adverse events) had a significantly higher (P = 0.019) incidence of adverse events than patients who had been taking 9 or fewer types of drugs per day (530 patients, of whom 100 had developed adverse events). The patients' attending physicians were not exclusively dementia specialists, they also included primary care physicians, circulatory medicine specialists, and gastroenterologists. There were patients who had also been prescribed various medications by physicians in multiple specialties, including orthopedic surgery and neurology, and information about the drugs that an individual patient was being prescribed was not necessarily being shared among all of the patient's doctors. It was confirmed that the prescribing of multiple drugs to patients receiving anti-dementia drugs made it more likely that these patients would develop adverse events, and this finding is consistent with the results of a study that was conducted previously in patients hospitalized at a single site [25].

Looking at patient treatment adherence when the patients were first visited at home by the pharmacists, it was found that patients who "forgot to take their medication 1–2 times a week" experienced significantly (P = 0.034) more adverse events than either patients who took their medication as directed or patients who hardly took their medication at all. This could be because when a patient resumes taking medication after the blood level of the drug has fallen because of a break in the treatment, the blood level either suddenly increases, or else the extent of the increase is greater than it would normally be, making it more likely that the patient will experience nausea or vomiting, or excitement or insomnia [32].

We asked the pharmacists to assess whether or not anti-dementia drugs were being used in patients for whom they are appropriate. The responses showed that in the 42 patients for whom the pharmacists determined that the use of anti-dementia drugs was inappropriate, 38.1% of the patients experienced adverse events, compared to an incidence of 19.5% in the group of 753 patients for whom the pharmacists determined that the use of anti-dementia drugs was appropriate (P = 0.009). In Japan, there are few options for the treatment of dementia other than drug therapy, and the treatment guidelines strongly recommend the use of anti-dementia drugs. This has resulted in anti-dementia dugs being used inappropriately in patients including elderly patients 85 years of age or older for whom there is little evidence to support the use of such therapy [36]. In the UK, the efficacy and applicability of the use of anti-dementia drugs has been contentious for a long time [16–18]. The National Institute for Health and Care Excellence (NICE) recommends that a patient's cognitive function, physical function, and functional/behavioral symptoms be assessed and reviewed regularly, and that therapy be continued only if the therapy is being found to afford efficacy [8]. This has helped to prevent the misuse of anti-dementia drugs and to promote the appropriate use of drug therapy.

This study had several limitations regarding participant recruitment and the assessments of the adverse events the patients were experiencing and the appropriateness of the drugs the patients were being prescribed. The first was that there may have been some bias in participant recruitment and data collection. Participant recruitment involved, as the first step, soliciting participation in the study from all of the members of the Japan Pharmaceutical Association by enclosing an invitation form in an issue of the Journal of the Japan Pharmaceutical Association. The pharmacists who participated in this study therefore likely constitute a pharmacist subpopulation that has a relatively high level of clinical expertise and a relatively high level of interest in anti-dementia drugs, and this means that some selection bias may be present. In addition, this study was not a typical retrospective survey; participants received the following instructions: "Please respond to the following questions about the patients to whom you are currently providing home visitation services. Please provide information on up to 5 such patients, regardless of whether or not they are taking anti-dementia drugs." In other words, the questions focused on collecting information about the current symptoms, prescriptions, treatment adherence, etc. of the patients. Pharmacists were not asked for information about, for example, events that occurred in the past or about patients' previous illnesses. Therefore, we think that there is virtually no recollection bias in the responses that were received. Secondly, as far as the pharmacists' understanding of the adverse events is concerned, although the responding pharmacists were not pharmacists who specialize in anti-dementia drug therapy, because the adverse events that occurred are not events that require a high degree of specialization to diagnose, we think that on the whole the current understanding of the patients' adverse events is accurate. However, we are not able to conclusively rule out the potential involvement in these adverse events of anti-dementia drugs or other factors (e.g., existing psychiatric or gastrointestinal illness, Meniere's disease, other lifestyle or stress factors). Thirdly, the questionnaire defined "inappropriate anti-dementia drug use" as cases in which either "the physician continued to prescribe an anti-dementia drug even though the patient had severe

dementia for which the anti-dementia drug was not indicated" or "the physician continued to prescribe an anti-dementia drug despite the fact that the therapeutic efficacy of the drug was not being regularly assessed and clear efficacy was not being achieved." The pharmacists responded to the survey questions based on these definitions. In Japan, most pharmacist home visitation services are provided based on the instructions of a primary care physician, and may include the prescribing of anti-dementia drugs. Japanese pharmacist home visitation services also include cases where both the physician and the pharmacist visit the patient together. It should be noted that, in Japan, neurologists specializing in dementia work full-time at their offices, and do not make home visits. Because there are no standardized assessment tools or criteria that are used by pharmacists to allow for a more rigorous assessment of "inappropriate anti-dementia drug use," such assessments vary depending on factors including the pharma-cist's ability and experience, and there consequently may have been some limitations in terms of the accuracy and consistency of these assessments.

The objective of this study was not to gain an understanding of what kinds of adverse events are caused by specific anti-dementia drugs or drug interactions. This study was conducted using a method that was different from conventional approaches in that it used data obtained directly from pharmacists, instead of, for example, the analysis of data from existing EHRs or physician reports. The use of anti-dementia drugs took off as soon as they became commer-cially available. They are now prescribed too readily, assessment of their efficacy is often per-functory, and adverse events are often overlooked. We investigated the prevalence of and risk factors for adverse events caused by anti-dementia drugs using a unique method, although this method may have some limitations. We hope the findings from our study will help improve the safety and efficacy of anti-dementia drugs in patients for whom they are prescribed.

## Author Contributions

**Conceptualization:** Hirohisa Imai, Takuya Hirai, Ryosuke Kumazawa, Shunsaku Nakagawa, Hiroyuki Nakao.

**Data curation:** Hirohisa Imai, Takuya Hirai, Ryosuke Kumazawa, Shunsaku Nakagawa, Atsushi Yonezawa, Kazuo Matsubara, Hiroyuki Nakao.

**Formal analysis:** Hirohisa Imai, Takuya Hirai, Ryosuke Kumazawa, Shunsaku Nakagawa, Atsushi Yonezawa, Kazuo Matsubara, Hiroyuki Nakao.

**Investigation:** Hirohisa Imai, Shunsaku Nakagawa, Atsushi Yonezawa, Kazuo Matsubara, Hir-oyuki Nakao.

**Methodology:** Ryosuke Kumazawa, Atsushi Yonezawa, Kazuo Matsubara, Hiroyuki Nakao.

**Project administration:** Hirohisa Imai, Shunsaku Nakagawa, Hiroyuki Nakao.

**Software:** Takuya Hirai, Kazuo Matsubara, Hiroyuki Nakao.

**Supervision:** Hirohisa Imai.

**Validation:** Hirohisa Imai, Atsushi Yonezawa, Kazuo Matsubara, Hiroyuki Nakao.

**Visualization:** Hirohisa Imai, Ryosuke Kumazawa, Shunsaku Nakagawa, Atsushi Yonezawa, Hiroyuki Nakao.

**Writing – original draft:** Hirohisa Imai, Hiroyuki Nakao.

**Writing – review & editing:** Hirohisa Imai, Takuya Hirai, Ryosuke Kumazawa, Shunsaku Nakagawa, Atsushi Yonezawa, Kazuo Matsubara, Hiroyuki Nakao.

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
