## [Decision Letter · Decision Letter 0]

9 Jan 2020

PONE-D-19-30063

Prevalence of and risk factors for adverse events in Alzheimer’s patients receiving anti-dementia drugs in at-home care

PLOS ONE

Dear Dr. Imai,

Thank you for submitting your manuscript to PLOS ONE. After careful consideration, we feel that it has merit but does not fully meet PLOS ONE’s publication criteria as it currently stands. Therefore, we invite you to submit a revised version of the manuscript that addresses the points raised during the review process.

Please, be aware that submitting a revision does not guarantee acceptance.

We would appreciate receiving your revised manuscript by Feb 23 2020 11:59PM. To enhance the reproducibility of your results, we recommend that if applicable you deposit your laboratory protocols in protocols.io, where a protocol can be assigned its own identifier (DOI) such that it can be cited independently in the future. For instructions see: http://journals.plos.org/plosone/s/submission-guidelines#loc-laboratory-protocols

We look forward to receiving your revised manuscript.

Kind regards,

Vincenzo De Luca

Academic Editor

PLOS ONE

Journal Requirements:

2. Please include additional information regarding the survey or questionnaire used in the study and ensure that you have provided sufficient details that others could replicate the analyses.

For instance, if you developed a questionnaire as part of this study and it is not under a copyright more restrictive than CC-BY, please include a copy, in both the original language and English, as Supporting Information.

3. In the Methods section, please clarify whether the IRB waived the need for patient consent, whether authors did not have access to identifying patient information, and whether the pharmacists gave consent to participate to the study.

Reviewers' comments:

Reviewer's Responses to Questions

**Comments to the Author**

1. Is the manuscript technically sound, and do the data support the conclusions?

Reviewer #1: Yes

Reviewer #2: Partly

2. Has the statistical analysis been performed appropriately and rigorously? 

Reviewer #1: Yes

Reviewer #2: Yes

3. Have the authors made all data underlying the findings in their manuscript fully available?

Reviewer #1: No

Reviewer #2: Yes

4. Is the manuscript presented in an intelligible fashion and written in standard English?

Reviewer #1: Yes

Reviewer #2: No

5. Review Comments to the Author

Reviewer #1: This study aims to identify types, prevalence, and risk factors of adverse events for patients receiving anti-dementia drugs in Japan.

A national wide survey was used to collect the information from pharmacists. The author found that 1) the most common type of adverse event is excitation/anxiety; 2) 21% of prevalence; 3) polypharmacy, inappropriate use, and irregular medication use are identified as potential risk factors for the adverse events.

This is an interesting study. The main limitation about this study is data collection and data harmonization. And the author acknowledges this limitation in this discussion part.

My main questions are:

1) How many dementia patients pharmacists typically deal with in this survey? Because this survey is retrospective, the accuracy about this questionnaire completely depends on the pharmacists’ memory and experience.

2) Are those pharmacists specialized in dementia symptoms only?

3) Can you also differentiate the two columns of p value in Table 4?

Reviewer #2: The study presented by the Authors is a survey based on pharmacists' responses to a questionnaire on anti-dementia drugs.

The only quantitative data the Authors found were that: 1) 21% of patients taking anti-dementia drugs showed adverse events of any kind; 2) the most relevant factors in predicting adverse events were: taking more than 10 drugs contemporarily and inappropriate use of anti-dementia drugs (as judged by the pharmacists).

These data appear too poor to grant their publication in Plos One and should better suit on a journal with less impact factor, eg a regional journal, also given their local resonance.

Most of all, these data were compromised by multiple methodological flaws:

1) the main goal of the study was not clearly stated. Authors should specify the aim of the study and should also indicate the outcome measure they wanted to examine (eg. rate of adverse events)

2) recruitment of respondents and data collection exposed to the risk of multiple biases, mostly selection and recalling bias, and the Authors did not mention how they managed them. Neither they discussed these possible biases in the Limitation section

3) quantitative analyses (i.e. those requiring statistical procedures) were only a minimal part of the manuscript, while the great part of it reported descriptive data, which are of limited scientific interest

4) no consideration was given to the possibility that multiple mediator factors may explain the association between exposure to anti-dementia drugs and adverse events.

4a) Taking 9-10 concurrent drugs with anti-dementia drugs was one significant risk factor of adverse events. How can it be excluded that the adverse events were caused by one of the 9-10 concurrent drugs or (even more plausible) by the pharmacodynamics/pharmacokinetics of these associations?

4b) What does it mean "inappropriate use" of anti-dementia drugs? How can pharmacists have all the information to judge upon the appropriateness of anti-dementia agents' prescription? Were the anti-dementia agents prescribed by primary-care physicians or by dementia specialists?

Given all these methodological and conceptual issues, together with the low relevance of data, in my opinion, this manuscript is not suitable for publication in Plos One

6. PLOS authors have the option to publish the peer review history of their article (what does this mean?). If published, this will include your full peer review and any attached files.

Reviewer #1: Yes: Yun Wang

Reviewer #2: No

---

## [Author Response · Author response to Decision Letter 0]

19 Feb 2020

PONE-D-19-30063

Reviewer #1: This study aims to identify types, prevalence, and risk factors of adverse events for patients receiving anti-dementia drugs in Japan.

A national wide survey was used to collect the information from pharmacists. The author found that 1) the most common type of adverse event is excitation/anxiety; 2) 21% of prevalence; 3) polypharmacy, inappropriate use, and irregular medication use are identified as potential risk factors for the adverse events.

This is an interesting study. The main limitation about this study is data collection and data harmonization. And the author acknowledges this limitation in this discussion part.

My main questions are:

1) How many dementia patients pharmacists typically deal with in this survey? Because this survey is retrospective, the accuracy about this questionnaire completely depends on the pharmacists’ memory and experience.

2) Are those pharmacists specialized in dementia symptoms only?

3) Can you also differentiate the two columns of p value in Table 4?

Response to Reviewer#1:

We are grateful for your careful review of our manuscript. Below, we will respond to your questions about our manuscript, and provide a supplemental explanation of the procedures that were used for data collection, harmonization, etc.

The method used to recruit respondents for this study was largely the same as the method that was used in a preceding study for which I, the lead author of this paper, was the principal investigator (BMJ Open. 2015 Aug 10;5(8):e007581. doi: 10.1136/bmjopen-2015-007581. PMID: 26260347. Correspondence to Hirohisa Imai); the only difference was the use of a paper questionnaire versus the use of a web-based questionnaire. Please refer to this aforementioned report.

Respondents were recruited in 2 stages. In the first stage, with the cooperation of the Japan Pharmaceutical Association, an invitation form soliciting participants for this study was enclosed in an issue of the Journal of the Japan Pharmaceutical Association, which is sent to all of the association’s members. In the second stage, the managing pharmacist of the pharmacy (the pharmacy manager) was designated as the person who would complete the questionnaire, and this pharmacy manager then accessed our web site and responded to the question items. We asked the responding pharmacists to enter into the questionnaire the information for up to 5 patients to whom the pharmacists had been providing home visitation services, regardless of whether or not the patients had been receiving anti-dementia drugs. The pharmacists were asked to enter information about the home visitation services that they personally had been providing.

Furthermore, we would like to point out that we set “up to 5” as the number of patients for whom information should be reported on the basis of the recruitment results of a pilot survey that had been conducted prior to this nationwide survey. Another reason was to reduce the burden placed on the pharmacists and thereby ensure a better response rate, and also to eliminate the effects of marked variations among the pharmacies in the number of patients.

1) How many dementia patients pharmacists typically deal with in this survey? Because this survey is retrospective, the accuracy about this questionnaire completely depends on the pharmacists’ memory and experience.

Response:

1) Of the patients whose cases were being handled by each pharmacist and out of whom up to 5 were selected at the pharmacist’s discretion, on average, 0.76 out of 5 patients were taking anti-dementia drugs. Although there were some pharmacists who only submitted information on 1 patient, out of all of the patients for whom information was submitted by 1136 pharmacists, there were 863 patients who were taking anti-dementia drugs, and the aforementioned average figure of 0.76 was calculated by dividing the number of patients taking anti-dementia drugs (863) by the total number of reporting pharmacists (1136).

The responding pharmacists were given the following instructions regarding this questionnaire: “Please respond to the following questions about the patients to whom you are currently providing home visitation services”; and “Please provide information on up to 5 such patients, regardless of whether or not they are taking anti-dementia drugs.” This study was therefore not a typical retrospective study, as we focused on questions about the “current” symptoms, medications, treatment adherence status, etc. of the patients. We did not ask for information about past illnesses or treatments. We therefore think that there is virtually no recollection bias in the responses that were received.

2) Are those pharmacists specialized in dementia symptoms only?

Response:

2) The responding pharmacists were not pharmacists who specialize in anti-dementia drug therapy. Therefore, the correctness of the decisions about the causal relationships between an adverse event had and an anti-dementia drug or about whether or not an anti-dementia drug was being used appropriately may, as was pointed out by Reviewer #1, vary somewhat depending on the pharmacist’s ability and experience. As is explained in our manuscript, because the pharmacists did not, for example, use any assessment tools, I think that the levels of objectivity and accuracy were not as high as they could have been. Although we stated in the manuscript that “because these adverse events were not symptoms that would require a great deal of expertise to diagnosis . . . we concluded that the patients’ adverse events had generally been assessed accurately,” we have rewritten this sentence to read as follows: “. . . because these adverse events were not events that require a high degree of specialization to diagnose, we think that on the whole the current understanding of the patients’ adverse events is accurate. However, we are not able to conclusively rule out the potential involvement in these adverse events of anti-dementia drugs or other factors (e.g., existing psychiatric or gastrointestinal illness, Meniere’s disease. or other lifestyle or stress factors).

3) We understand your request to explain the difference in the P values on the left and right of Table 4. The P values on the right are the values for a multivariate analysis; we have noted this by putting a superscript “a” next to “P value” at the top of the right-hand column.

Reviewer #2: The study presented by the Authors is a survey based on pharmacists' responses to a questionnaire on anti-dementia drugs.

The only quantitative data the Authors found were that: 1) 21% of patients taking anti-dementia drugs showed adverse events of any kind; 2) the most relevant factors in predicting adverse events were: taking more than 10 drugs contemporarily and inappropriate use of anti-dementia drugs (as judged by the pharmacists).

These data appear too poor to grant their publication in Plos One and should better suit on a journal with less impact factor, eg a regional journal, also given their local resonance.

Most of all, these data were compromised by multiple methodological flaws:

Response to Reviewer#2:

We appreciate your review, although it was critical of our manuscript. It appears to us that you found our initially submitted manuscript insufficiently convincing because our explanations were inadequate and/or lacking in clarity. We would like to provide a more complete explanation showing that our research yielded findings that can help to solve the problems that are associated with anti-dementia drugs and is therefore worthy of being published in PLOS One.

The method used to recruit respondents for this study was generally the same as the method that was used in a preceding study for which I, the lead author of this paper, was the principal investigator (BMJ Open. 2015 Aug 10;5(8):e007581. doi: 10.1136/bmjopen-2015-007581. PMID: 26260347. Correspondence to Hirohisa Imai); the only difference was the use of a paper questionnaire versus the use of a web-based questionnaire. Please refer to this aforementioned report.

Respondents were recruited in 2 stages. In the first stage, with the cooperation of the Japan Pharmaceutical Association, an invitation form soliciting participants for this study was enclosed in an issue of the Journal of the Japan Pharmaceutical Association, which is sent to all of the association’s members. In the second stage, the managing pharmacist (pharmacy manager) of the pharmacy was designated as the person who would complete the questionnaire, and this pharmacy manager then accessed our web site and responded to the question items. We asked the responding pharmacists to enter into the questionnaire the information for up to 5 patients to whom the pharmacists had been providing home visitation services, regardless of whether or not the patients had been receiving anti-dementia drugs. The pharmacists were asked to enter into the questionnaire information about the home visitation services that they personally had been providing.

Furthermore, we would like to point out that we set “up to 5” as the number of patients for whom information should be reported on the basis of the recruitment results of a pilot survey that had been conducted prior to this nationwide survey. Another reason was to reduce the burden placed on the pharmacists and thereby ensure a better response rate, and also to eliminate the effects of marked variations among the pharmacies in the number of patients.

1) the main goal of the study was not clearly stated. Authors should specify the aim of the study and should also indicate the outcome measure they wanted to examine (eg. rate of adverse events)

Response:

We have revised the description of the study’s objectives to read as follows:

“The objective of this study is to clarify the types and frequencies of, and risk factors for, the adverse events that occur in patients taking anti-dementia drugs.”

We have also added the following sentence to the “Results” section: “The overall rate of adverse events was 20.9%.”

2) recruitment of respondents and data collection exposed to the risk of multiple biases, mostly selection and recalling bias, and the Authors did not mention how they managed them. Neither they discussed these possible biases in the Limitation section

Response:

This study was not a typical retrospective survey; participants received the following instructions: “Please respond to the following questions about the patients to whom you are currently providing home visitation services. Please provide information on up to 5 such patients, regardless of whether or not they are taking anti-dementia drugs.” In other words, the questions focused on collecting information about the current symptoms, prescriptions, treatment adherence, etc. of the patients. Pharmacists were not asked for information about, for example, events that occurred in the past or about patients’ previous illnesses. Therefore, we think that there is virtually no recollection bias in the responses that were received. Regarding selection bias, the pharmacists who participated in this study likely constitute a pharmacist subpopulation that has a relatively high level of interest in anti-dementia drugs. We did not implement any measures for avoiding this selection bias when we recruited study participants or performed the data analysis. Some selection bias may therefore be present.

We have revised our description of the limitations in the “Discussion” section to read as follows: “We solicited participation in this study from all of the members of the Japan Pharmaceutical Association through an invitation form that was enclosed in an issue of the Journal of the Japan Pharmaceutical Association. The pharmacists who participated in this study likely constitute a pharmacist subpopulation that has a relatively high level of clinical expertise and a relatively high level of interest in anti-dementia drugs. Some selection bias may therefore be present.”

3) quantitative analyses (i.e. those requiring statistical procedures) were only a minimal part of the manuscript, while the great part of it reported descriptive data, which are of limited scientific interest

Response:

Aside from performing the multivariate logistic regression analysis presented in Table 4, we did not apply any other higher-order statistical procedures. However, we think that we have appropriately described the fundamental data, and that the chi-squared, Fisher’s exact, and trend tests have been appropriately utilized and are appropriate given the nature of the data. These basic statistical procedures required few assumptions compared to higher-order statistical procedures, and constituted a methodology that was robust with respect to deviations from the assumptions. There is consequently no need for the reader to be concerned about checking whether or not the required assumptions were satisfied, and because the statistical procedure was a fundamental procedure, it is easy to understand and ensures a high degree of analytical transparency. We therefore think that it is an approach that is not inferior to higher-order statistical procedures.

4) no consideration was given to the possibility that multiple mediator factors may explain the association between exposure to anti-dementia drugs and adverse events.

4a) Taking 9-10 concurrent drugs with anti-dementia drugs was one significant risk factor of adverse events. How can it be excluded that the adverse events were caused by one of the 9-10 concurrent drugs or (even more plausible) by the pharmacodynamics/pharmacokinetics of these associations?

Response:

As you have pointed out, it is not clear what drugs or drug interactions caused the adverse events. One of the important findings of our study is that the probability of adverse events occurring increases as the number of concomitant medications increases. In general, polypharmacy is common in dementia patients, and our findings are significant from the standpoint of ensuring the safety of drug treatment. The objective of this study was to investigate risk factors, with the emphasis not on “pharmacological factors, such as drug interactions and pharmacokinetics,” but rather “sociopharmaceutical factors, such as the characteristics and treatment adherence of individual patients.”

4b) What does it mean "inappropriate use" of anti-dementia drugs? How can pharmacists have all the information to judge upon the appropriateness of anti-dementia agents' prescription? Were the anti-dementia agents prescribed by primary-care physicians or by dementia specialists?

Response:

The questionnaire defined “inappropriate anti-dementia drug use” as cases in which either “the physician continued to prescribe an anti-dementia drug even though the patient had severe dementia for which the anti-dementia drug was not indicated” or “the physician continued to prescribe an anti-dementia drug despite the fact that the therapeutic efficacy was not being regularly assessed and clear efficacy was not being achieved.”

The pharmacists responded to the survey questions based on these definitions. In Japan, pharmacist home visitation services sometimes include cases where both the physician and the pharmacist visit the patient together. In general, pharmacists who provide home visitation services acquire a better understanding of their patients’ clinical conditions than pharmacists who remain at the pharmacy.

Anti-dementia drugs are prescribed by primary care physicians. Pharmacist home visitation services are mostly provided pursuant to the instructions of primary care physicians. In Japan, physicians specializing in dementia work full-time at their offices, and do not make home visits.

As noted by Reviewer #2, there may be some limitations to this study regarding the accuracy of the pharmacists’ assessments of adverse events and inappropriate use. We have therefore revised the description of the study’s limitations in the “Discussion” section.

---

## [Decision Letter · Decision Letter 1]

19 Mar 2020

Prevalence of and risk factors for adverse events in Alzheimer’s patients receiving anti-dementia drugs in at-home care

PONE-D-19-30063R1

Dear Dr. Imai,

We are pleased to inform you that your manuscript has been judged scientifically suitable for publication and will be formally accepted for publication once it complies with all outstanding technical requirements.

With kind regards,

Vincenzo De Luca

Academic Editor

PLOS ONE

Additional Editor Comments (optional):

Reviewers' comments:

Reviewer's Responses to Questions

**Comments to the Author**

1. If the authors have adequately addressed your comments raised in a previous round of review and you feel that this manuscript is now acceptable for publication, you may indicate that here to bypass the “Comments to the Author” section, enter your conflict of interest statement in the “Confidential to Editor” section, and submit your "Accept" recommendation.

Reviewer #2: (No Response)

2. Is the manuscript technically sound, and do the data support the conclusions?

Reviewer #2: Yes

3. Has the statistical analysis been performed appropriately and rigorously? 

Reviewer #2: Yes

4. Have the authors made all data underlying the findings in their manuscript fully available?

Reviewer #2: Yes

5. Is the manuscript presented in an intelligible fashion and written in standard English?

Reviewer #2: Yes

6. Review Comments to the Author

Reviewer #2: The Authors clarified the points of major criticism I rasied in my previous review. I feel satisfied.

7. PLOS authors have the option to publish the peer review history of their article (what does this mean?). If published, this will include your full peer review and any attached files.

Reviewer #2: No

---

## [Editor Report · Acceptance letter]

23 Mar 2020

PONE-D-19-30063R1 

Prevalence of and risk factors for adverse events in Alzheimer’s patients receiving anti-dementia drugs in at-home care 

Dear Dr. Imai:

I am pleased to inform you that your manuscript has been deemed suitable for publication in PLOS ONE. Congratulations! Your manuscript is now with our production department. 

With kind regards,

on behalf of

Dr. Vincenzo De Luca 

Academic Editor

PLOS ONE